# Auditory selective attention is enhanced by a task-irrelevant temporally coherent visual stimulus in human listeners

**Ross K Maddox[1], Huriye Atilgan[2], Jennifer K Bizley[2], Adrian KC Lee[1,3]\***

[1]Institute for Learning and Brain Sciences, University of Washington, Seattle, United States; [2]Ear Institute, University College London, London, United Kingdom; [3]Department of Speech and Hearing Sciences, University of Washington, Seattle, United States

**Abstract** In noisy settings, listening is aided by correlated dynamic visual cues gleaned from a talker's face—an improvement often attributed to visually reinforced linguistic information. In this study, we aimed to test the effect of audio–visual temporal coherence alone on selective listening, free of linguistic confounds. We presented listeners with competing auditory streams whose amplitude varied independently and a visual stimulus with varying radius, while manipulating the cross-modal temporal relationships. Performance improved when the auditory target's timecourse matched that of the visual stimulus. The fact that the coherence was between task-irrelevant stimulus features suggests that the observed improvement stemmed from the integration of auditory and visual streams into cross-modal objects, enabling listeners to better attend the target. These findings suggest that in everyday conditions, where listeners can often see the source of a sound, temporal cues provided by vision can help listeners to select one sound source from a mixture.

*For correspondence: akclee@uw.edu

Competing interests: The authors declare that no competing interests exist.

## Introduction

A key challenge faced by the auditory system is to appropriately segregate and group sound elements into their component sources. This process of auditory scene analysis underlies our ability to listen to one sound while ignoring others and is crucial for everyday listening. While psychophysical studies have focused on the perceptual 'rules' that govern the likelihood of elements being grouped into single objects in audition (*Bregman, 1990*; *Bizley and Cohen, 2013*) or in vision (*Marr, 1982*; *Lee and Yuille, 2006*), perception is a seamlessly multisensory process whereby sensory information is integrated both within and across different sensory modalities. It is of great interest to understand how the formation of cross-modal objects might influence perception, and in particular, whether visual information may provide additional cues that listeners can use to facilitate auditory scene segregation through the generation of auditory-visual objects. To date, there has yet to be a study utilizing ongoing, continuous stimuli that demonstrates a perceptual benefit of an uninformative visual stimulus when selectively attending to one auditory stimulus in a mixture.

Many previous studies have focused on audio–visual speech processing. When listening in a noisy setting, watching a talker's face drastically improves speech intelligibility (*Binnie, 1973*; *Bernstein and Grant, 2009*) and this benefit is particularly apparent under difficult listening conditions, such as when speech is masked by spatially coincident competing speech (*Helfer and Freyman, 2005*). Much work has focused on the role that visual information plays in contributing linguistic information—for example, the unvoiced consonants /p/, /t/, and /k/ can be disambiguated based upon mouth movements, a fact which underlies the McGurk effect whereby conflicting acoustic and visual information create an intermediary percept, changing which syllable a listener hears (*McGurk and MacDonald, 1976*).

**eLife digest** In the noisy din of a cocktail party, there are many sources of sound that compete for our attention. Even so, we can easily block out the noise and focus on a conversation, especially when we are talking to someone in front of us.

This is possible in part because our sensory system combines inputs from our senses. Scientists have proposed that our perception is stronger when we can hear and see something at the same time, as opposed to just being able to hear it. For example, if we tried to talk to someone on a phone during a cocktail party, the background noise would probably drown out the conversation. However, when we can see the person we are talking to, it is easier to hold a conversation.

Maddox et al. have now explored this phenomenon in experiments that involved human subjects listening to an audio stream that was masked by background sound. While listening, the subjects also watched completely irrelevant videos that moved in sync with either the audio stream or with the background sound. The subjects then had to perform a task that involved pushing a button when they heard random changes (such as subtle changes in tone or pitch) in the audio stream.

The experiment showed that the subjects performed well when they saw a video that was in sync with the audio stream. However, their performance dropped when the video was in sync with the background sound. This suggests that when we hold a conversation during a noisy cocktail party, seeing the other person's face move as they talk creates a combined audio–visual impression of that person, helping us separate what they are saying from all the noise in the background. However, if we turn to look at other guests, we become distracted and the conversation may become lost.

Such benefits are specific to speech signals, however, leaving questions about the general principles that govern multisensory integration and its perceptual benefits.

In the case of selective listening, visual timing cues may provide important listening benefits. When performing simple tone-in-noise detection, thresholds improve when the timing of the potential tone is unambiguous (*Watson and Nichols, 1976*). Visual speech cues, which precede the auditory signal (*Chandrasekaran et al., 2009*; *Stevenson et al., 2012*), provide helpful information about when to listen (*Grant and Seitz, 2000*; *Grant, 2001*; *Bernstein et al., 2004*) and may help target speech to be separated from other competing speech (*Summerfield, 1992*; *Devergie et al., 2011*). In addition to information about when to listen, timing may be important for another reason: temporal coherence may facilitate binding of auditory and visual stimuli into single cross-modal objects. However, previous studies of audio–visual binding—the integration of auditory and visual stimuli into a single percept—have focused on judgments of coherence or simultaneity (*Recanzone, 2003*; *Spence and Squire, 2003*; *Fujisaki and Nishida, 2005*; *Denison et al., 2013*), rather than on the perceptual enhancements provided by that binding.

Past studies employing transient stimuli have shown a number of ways in which task-irrelevant stimuli in one modality can affect perception of another modality. A sound can affect the perceived number of visual stimuli (*Shams et al., 2000*, *2002*; *Bizley et al., 2012*), the color (*Mishra et al., 2013*), and the direction of visual motion (*Freeman and Driver, 2008*). A task-irrelevant sound can also increase detection of a visual stimulus (*McDonald et al., 2000*) by affecting the bias and sensitivity, as well as alter the perceived visual intensity (*Stein et al., 1996*; *Odgaard et al., 2003*). Similarly, an irrelevant visual stimulus can affect the detection of an auditory stimulus (*Lovelace et al., 2003*) as well as the perceived loudness (*Odgaard et al., 2004*). These effects could all conceivably help when segregating streams, but none of these studies demonstrated such benefits.

In this study, we developed a novel paradigm that was designed to test whether cross-modal temporal coherence was sufficient to promote the segregation of two competing sound sources. To achieve this, we created audio–visual stimuli with dynamic, continuous noise envelopes. We chose to manipulate temporal coherence in an ethologically relevant modulation frequency range (*Chandrasekaran et al., 2009*), amplitude modulating our stimuli with a noisy envelope low-pass filtered at 7 Hz. Listeners were asked to perform an auditory selective attention task that required them to report brief perturbations in the pitch or the timbre of one of two competing streams. Additionally, a concurrent visual stream was presented where the radius of the visual stimulus could vary coherently with the amplitude changes either of the target or the masker, or be independent of both. We hypothesized that modulating the

visual stimulus coherently with one of the auditory streams would cause these stimuli to automatically bind together, with a consequent improvement in performance when the visual stimulus was coherent with the target auditory stream. Importantly, any perceptual benefit could only result from the temporal coherence between the auditory and visual stimulus, as the visual stimulus itself provided no additional information about the timing of the target auditory perturbations.

## Results

All methods were approved by the Institutional Review Board of the University of Washington and the Ethics Committee of the University College London (ref: 5139). We sought to measure a behavioral benefit that could be ascribed directly to the temporal coherence between auditory and visual streams: specifically, whether coherence between a visual stimulus and a task-irrelevant auditory feature could improve performance in an auditory selective attention task. Importantly, the visual stimulus was uninformative of the auditory task, such that any behavioral benefit could be unambiguously determined to be the result of selective enhancement of an auditory stream due to binding with the visual stimulus. This cross-feature enhancement thus served simultaneously as our assay of audio–visual binding (eschewing subjective judgments) and an investigation of binding's impact on auditory scene analysis.

We designed a task that required selectively attending to one of two competing auditory streams and manipulated the coherence of the visual stimulus relative to that of the target and masker auditory streams. The temporal dynamics of each stream were defined by low-pass noise envelopes with a 7 Hz cutoff frequency, roughly approximating the modulation frequency range of speech amplitude envelopes and visual mouth movements (*Chandrasekaran et al., 2009*). In each trial of 14 s duration there were two auditory streams (one target, one masker; *Figure 1A*) that were amplitude-modulated by independent tokens of the noise envelope and one visual stream: a gray disc surrounded by a white ring whose radius was also modulated by a noise envelope (envelope in *Figure 1B*; images of disc in *Figure 1C*). The visual radius envelope could match the amplitude envelope of the target auditory stream (example stimuli in *Video 1*) or masker auditory stream (*Video 2*), or be independent from both (*Video 3*). This audio–visual coherence (specifically which, if either, auditory stream's amplitude envelope was matched by the visual radius) defined the three experimental conditions, henceforth referred to as match-target, match-masker, and match-neither, respectively (*Figure 1B*). The goal of the subject was to respond by pressing a button to brief perturbation events in the target stream and ignore events in the masker stream. Importantly, the task-relevant auditory feature was not the same feature that was coherent with the visual stimulus. A response to an event in the target stream would be deemed a 'hit' and a response to a masker event would be deemed a 'false alarm', allowing us to calculate $d'$ sensitivity for each of the three conditions. Subjects were also required to respond to brief color change flashes in the visual stimulus (*Figure 1F*)—this task was not difficult (overall hit rate was 87.7%), but ensured attentiveness to the visual stream. We reason that an improvement in behavioral performance in the match-target condition over the match-masker condition could only result from a benefit in separating the target and masker stream and thus be indicative of true, ongoing cross-modal binding, beyond a simple subjective report.

If true cross-modal objects were being formed, then the auditory perceptual feature to which the listener must attend should not matter. To assess this generality, two types of auditory events were used, in two different groups of subjects. For half of the subjects (N = 16), each auditory stream was an amplitude-modulated tone (fundamental frequency, F0, of 440 or 565 Hz, counterbalanced), and the events were 100 ms fluctuations in the carrier frequency of ±1.5 semitones (*Figure 1D*). For the other half of subjects (N = 16), each auditory stream was a band-pass filtered click train with a distinct pitch and timbre (synthetic vowels /u/ and /a/; F0 of 175 and 195 Hz), and the events were small changes in the timbre of the stream generated by slightly modulating the first and second formant frequencies (F1 and F2, *Figure 1E*).

In addition to $d'$, we calculated response bias (calculated as ln $\beta$, where $\beta$ is the likelihood ratio of a present target vs an absent target at the subject's decision criterion [*Macmillan and Creelman, 2005*]), hit rates, false alarm rates, and visual hit rates. These values are all plotted in the left panels of *Figure 2*. Since there was broad variation in subjects' overall abilities, as well as in the difficulty between pitch and timbre tasks, the right panels of *Figure 2* show the across-subject means in each condition relative to each subject's overall mean. This is the visual parallel of the within-subjects statistical tests described below.

Subjects were more sensitive to target events when the visual stimulus was coherent with the target auditory stream than when it was coherent with the masker. We ran an ANOVA for $d'$, bias, hit rate,

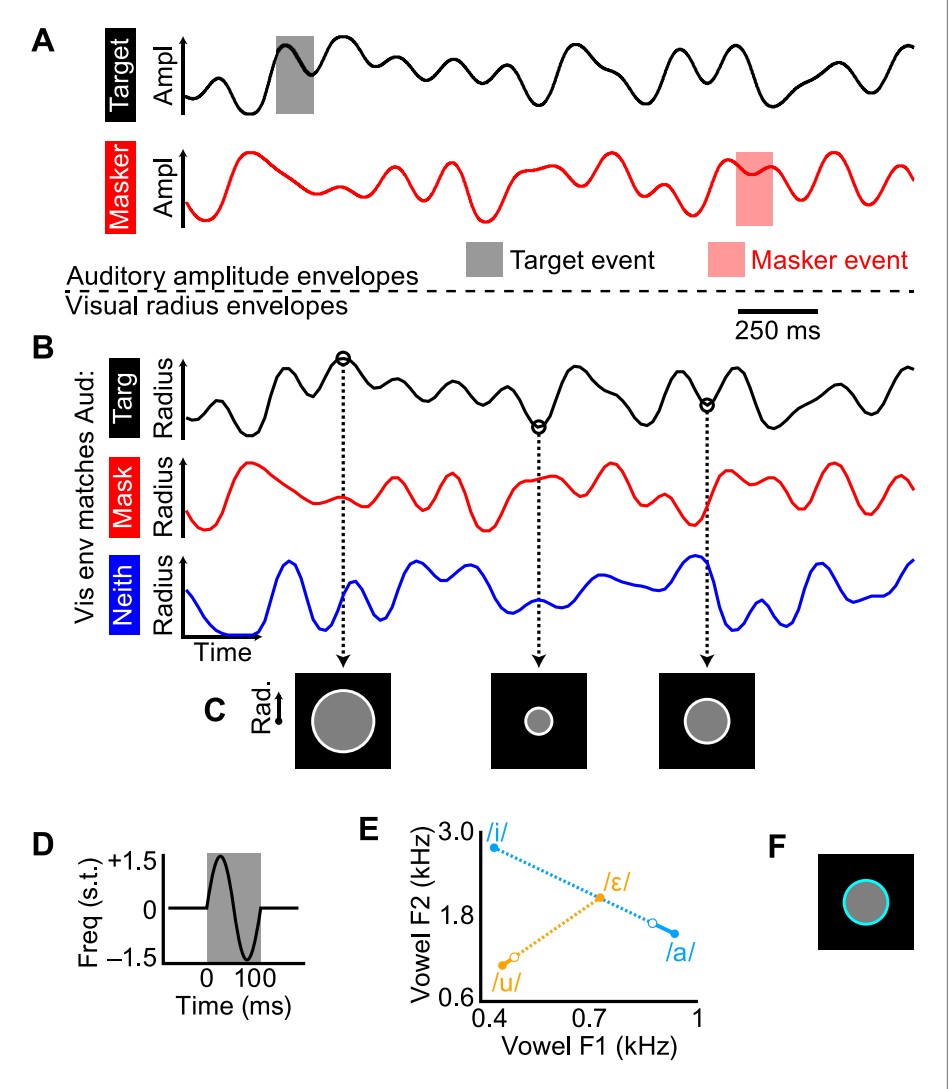

**Figure 1.** Construction of the auditory and visual stimuli. (**A**) Amplitude envelopes shown for 2 s of the target (black) and masker (red) auditory streams. Trials were 14 s long, over which the target and masker envelopes were independent. (**B**) Visual radius envelopes for the three audio–visual coherence conditions: match-target (black), match-masker (red), and match-neither (blue). (**C**) Example frames of the visual disc at three radius values, according to the match-target envelope in **B**. (**D**) Carrier frequency modulation events for the pitch task. Deflection was one period of a sinusoid, reaching ±1.5 semitones over 100 ms. (**E**) Changes in vowel formants F1 and F2 for the timbre events. There were two streams, one with vowel /u/ and the other with vowel /a/. Timbre events lasted 200 ms and morphed formants F1 and F2 slightly toward /ɛ/ and /i/, respectively, and then back to /u/ and /a/. The closed circle endpoints show the steady-state vowel and the open circle point shows the average reversal point across subjects. Note that the change in formats during the morph event was small compared to the distance between vowels in the F1–F2 space. (**F**) The visual stimulus during a flash (100 ms duration).

false alarm rate, and visual hit rate with a between-subjects factor of auditory event type (pitch or timbre) and a within-subjects factor of audio–visual coherence condition (match-target, match-masker, match-neither) on the data uploaded as *Figure 2—source data 1*. For *d'*, we found a significant between-groups effect of event type [F(1, 30) = 9.36, p = 0.0046, α = 0.05] and a significant within-subjects effect of coherence [F(2, 60) = 4.28, p = 0.018]. There was no interaction between these two factors (p = 0.60), indicating the generality of the effect of cross-modal coherence on the selective attention task to different features and experimental setups, as well as different task difficulties (as performance was significantly better with pitch events vs timbre events). *Post hoc* comparisons demonstrated that

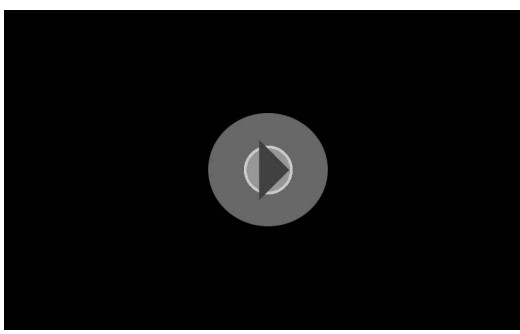

**Video 1**. Two example trials from the match-target condition. The video shows two trials from the pitch task in which the target auditory stream is coherent with the visual stimulus. The target auditory stream starts 1 s before the masker stream (lower pitch in the first trial, higher pitch in the second). The task is to respond by pressing a button to brief pitch perturbations in the target auditory stream but not the masker auditory stream, as well as to cyan flashes in the ring of the visual stimulus.

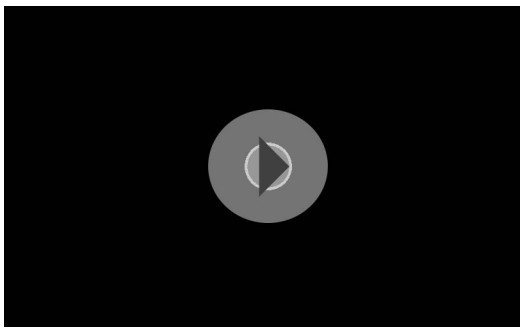

**Video 2**. Two example trials from the match-masker condition. As in **Video 1**, except the visual stream is coherent with the masker auditory stream.

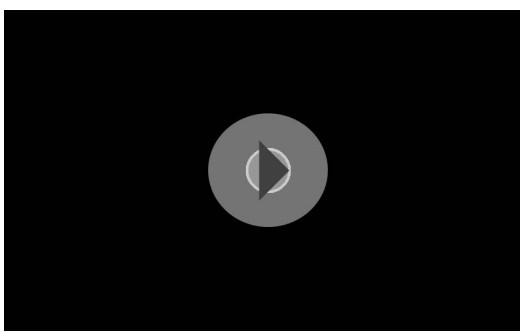

**Video 3**. Two example trials from the match-neither condition. As in **Video 1**, except the visual stream is coherent neither auditory stream.

subjects performed better when the visual stimulus was coherent with the target auditory stream vs the auditory masker (match-target > match-masker; p = 0.0049, Bonferroni-corrected α = 0.017). Similar results were obtained for hit rate: there was a significant effect of event type [F(1, 30) = 10.1, p = 0.0034] and also of visual coherence [F(2, 60) = 1.286, p = 0.0497]. *Post hoc* tests also showed a significant difference between hit rate in the match-target and match-masker conditions (p = 0.011). These results suggest that a visual stimulus can enhance a coherent auditory stimulus, even when that visual stimulus provides no useful visual information. There was no significant difference between the match-neither condition and either of the other two conditions. Thus, while there is a clear benefit when the visual stimulus is coherent with an attentional target vs a distractor, it is not yet clear if the changes in performance represent a helpful enhancement of the target or a deleterious enhancement of the masker vs some neutral condition, such as the match-neither case or an auditory-only case.

For bias, there was no significant effect of event type or coherence condition (ANOVA as above, p = 0.284 and 0.985, respectively), and subjects were generally conservative in their responses (ln $\beta$ > 0, intercept significant at F[1, 30] = 43.6, p < 0.0005). The lack of a variation in bias solidifies the notion that the observed pattern of responses results from changes in auditory detectability. Neither factor had a significant effect on false alarm rate (p = 0.29 and 0.18) or visual hit rate (p = 0.10 and 0.091). For this reason, *post hoc* paired comparisons were not made.

## Discussion

Here, we used temporal coherence between auditory and visual features to influence the way subjects parsed an audio–visual scene. Many studies have examined how the timing of single auditory and visual events (**Fujisaki and Nishida, 2005**; **Zampini et al., 2005**) or simple periodic modulations of stimulus features (**Recanzone, 2003**; **Spence and Squire, 2003**; **Fujisaki and Nishida, 2005**) influence the likelihood of an integrated multisensory percept. However, the temporal dynamics of one event or a repeating sequence are quite unlike many natural sounds, including speech. A recent study addressed this by creating randomly timed sequences of discrete auditory-visual events, and showed that coherence discrimination was better for the unpredictable sequences than for predictable ones (**Denison et al., 2013**). Thus, the noisy dynamics of stimulus features used here not only allowed

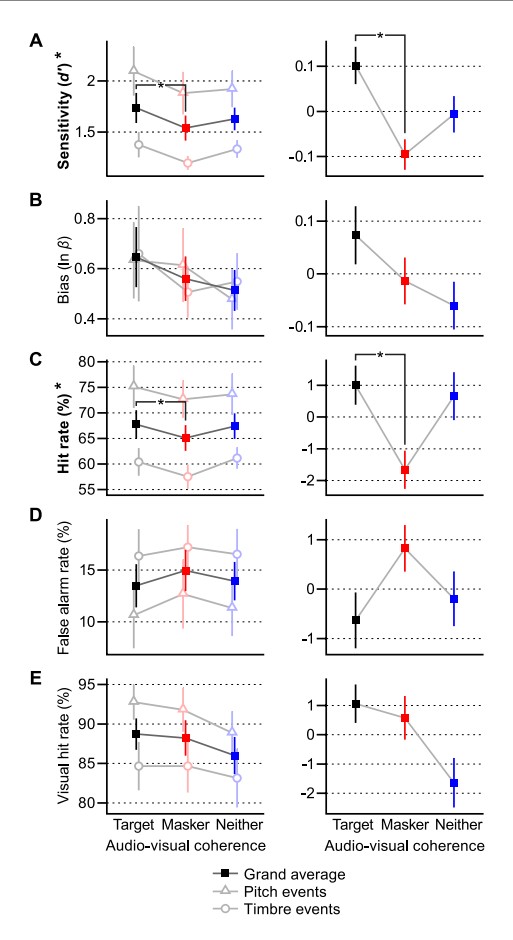

**Figure 2**. Behavioral results. Each behavioral measure shown in two panels: (**A**) *d′* sensitivity, (**B**) bias, (**C**) hit rate, (**D**) false alarm rate, (**E**) visual hit rate. Left: mean ± SEM for each condition across all subjects (solid squares) as well as for pitch and timbre events separately (empty triangles and circles, respectively). Right: normalized mean ± SEM across all subjects demonstrating the within-subjects effects. Measurements with significant effects of coherence (viz., sensitivity and hit rate) are denoted with bold type and an asterisk on their vertical axis label. *Post hoc* differences between conditions that are significant at p < 0.017 are shown with brackets and asterisks. See 'Results' for outcomes of all statistical tests.

The following source data is available for figure 2:

**Source data 1**. Behavioral results for individual subjects.

our stimuli to more closely emulate speech, but likely also strengthened the bound percept.

The integration of visual and auditory information to create a cross-modal percept relies on the sensory signals being bound together. The well-known McGurk illusion (*McGurk and MacDonald, 1976*) can be reduced by contextually 'unbinding' the auditory and visual stimuli by preceding a test trial with non-matching audio and video (*Nahorna et al., 2012*). Illusory syllables are also more likely when audio is paired with a dynamic face as opposed to a static image of a face that provides the same information regarding place of articulation (*Rosenblum and Saldaña, 1996*). Additionally, individual differences in purely temporal processing (specifically, how long a discrete auditory event can lag a visual event and still be perceived as simultaneous) are predictive of audio–visual fusion in multiple cross-modal illusions (*Stevenson et al., 2012*).

To our knowledge, the present study is the only one to base a behavioral task on one auditory feature and use an orthogonal feature to create binding with the visual stimulus, thus demonstrating an enhancement of an auditory feature about which the visual stimulus was uninformative. How, then, did a coherent visual stimulus facilitate enhanced behavioral performance?

Coherence-driven binding of auditory and visual features to create a cross-modal object may underlie the present results. In the unimodal case, temporal coherence between auditory features (i.e., pitch, timbre, location) is a principal contributor to the formation of auditory objects (*Shamma et al., 2011*; *Teki et al., 2013*); however, the definition of an object can be difficult to pin down. *Griffiths and Warren (2004)* suggest criteria for what constitutes an auditory object. They first suggest that the incoming information corresponds to something in the sensory world; this often means a physical sound source, but they leave room in their definition for artificial sounds that could conceivably come from a real-world source. Second, object-related information should be separable from other incoming sensory information—this was a major component of the present task. Third, an object demonstrates perceptual invariance. While this was not a focus of the present study, listeners' perceptions of the attended features were invariant in the face of fluctuating amplitude. The fourth and most interesting (in the context of the present discussion) criterion is that an object is generalizable between senses. The authors point out that it is 'less clear' whether analyzing sensory objects is affected by cross-modal correspondence. The notion of cross-modal objecthood is easy to accept but difficult to demonstrate conclusively. The sound-induced flash illusion (*Shams et al., 2000, 2002*), where the number of quick flashes perceived is strongly biased by the number of coincident beeps, is affected by audio–visual spatial congruence when observers are required to selectively attend to two spatially separated stimulus streams (*Bizley et al., 2012*). This suggests that cross-modal sensory

interaction is strongest when evidence supports information from both modalities as originating from the same object.

We prefer a functional definition of 'objecthood' because it provides testable predictions about the perception of a purported object's features. There is evidence from unimodal studies that both visual and auditory attention operate on perceptual objects, such that attention to one feature of an object enhances the perception of all its features (*Alain and Arnott, 2000*; *Blaser et al., 2000*; *Shinn-Cunningham, 2008*; *Shamma et al., 2011*; *Maddox and Shinn-Cunningham, 2012*). Here, we show that processing of auditory pitch and timbre are enhanced when auditory amplitude is coherent with visual size. In the case of the match-target condition, this is beneficial, with attention to the visual stimulus improving listening performance. Performance suffers in the match-masker case, where the incorrect auditory stream's features are enhanced through the same binding process.

*Figure 3* shows a conceptual model of the processing underlying these results, using the pitch task as an example. Auditory target and masker (on the left) and visual streams (on the right) are depicted as bound sets of component features, with those connections shown as line segments. When there is cross-modal temporal coherence between a feature of an auditory and visual stream, those features are bound. This results in a cross-modal object (just as two auditory or visual streams with the same envelope very likely would have bound together as well). In the match-target condition (*Figure 3A*), the to-be-attended features (highlighted with a yellow ellipse) become part of one object and are enhanced (shown by a thick bounding box). The features of the to-be-ignored stream are suppressed (shown as a broken bounding box). However, in the match-masker condition (center), attention to the target auditory and visual features is now across objects. To make matters worse, processing of the to-be-ignored auditory features is enhanced, increasing the false-alarm likelihood. In the match-neither case, attention must still be split between unbound auditory and visual streams, but the masking stream is not enhanced, leading to performance between the other two conditions. How this model might be biologically implemented remains to be elucidated, but one possibility is that cross-modal coherence between stimulus modalities enables neural activity patterns in visual cortex to reinforce the activity elicited by the coherent auditory stream in auditory cortex (*Ghazanfar and Schroeder, 2006*; *Bizley and King, 2012*).

If the formation of cross-modal objects underlies the present results, then the enhancement should be bidirectional. While the present experiment was designed specifically to test the effect of a visual stimulus on auditory performance, the model also makes predictions for enhancement of a task-irrelevant visual cue. Since there was only one visual stimulus (i.e., no competing distractor visual stimulus), its features should be enhanced when it is coherent with any auditory stimulus compared to when it is not, but further experiments specifically designed to test this bi-directionality are needed.

Within one modality, a stream or object can be thought of as occupying a point or region of a multidimensional feature space: it can have a specific pitch, location, onset or offset time, etc that define it (*Shamma et al., 2011*; *Teki et al., 2011, 2013*; *Micheyl et al., 2013*; *Christiansen et al., 2014*). Two streams may bind if they are close together in one or more of these dimensions, such as tones of two frequencies that start and stop at the same time (*Shamma et al., 2011*). Similar principles should govern the formation of cross-modal objects, with the caveat that only some stimulus features span sensory boundaries. These dimensions could be physical, such as time and space for audio-visual stimuli (*Alais and Burr, 2004*; *Talsma et al., 2010*; *Bizley et al., 2012*), or learned statistical correspondences, such as the linguistic connection between an image of someone with protruded lips and an auditory /u/ vowel. Within this object, all features are enhanced and disagreements or ambiguities are resolved, either by letting the more reliable modality dominate (typically vision for space, audition for timing [*Talsma et al., 2010*]), or by creating an intermediary percept (e.g., the McGurk effect). Such integration is necessary for a perceptual object to be a reasonable model of a physical one, which cannot, for example, occupy more than one location at one time.

How does one establish that cross-modal binding has occurred? Here, we suggest that the demonstration of feature enhancement within a putative object is the marker of binding. We showed that attending to the color of a visual stream enhanced the perception of the pitch or timbre of an auditory stream, despite the fact that those two (i.e., color and pitch or color and timbre) features on their own were unrelated. This influence instead occurred through a chain of bound features within a cross-modal object, with a shared temporal trajectory linking visual size and auditory amplitude and is, to our knowledge, the first example of a continuous visual signal benefitting listening abilities through enhanced auditory scene segregation.

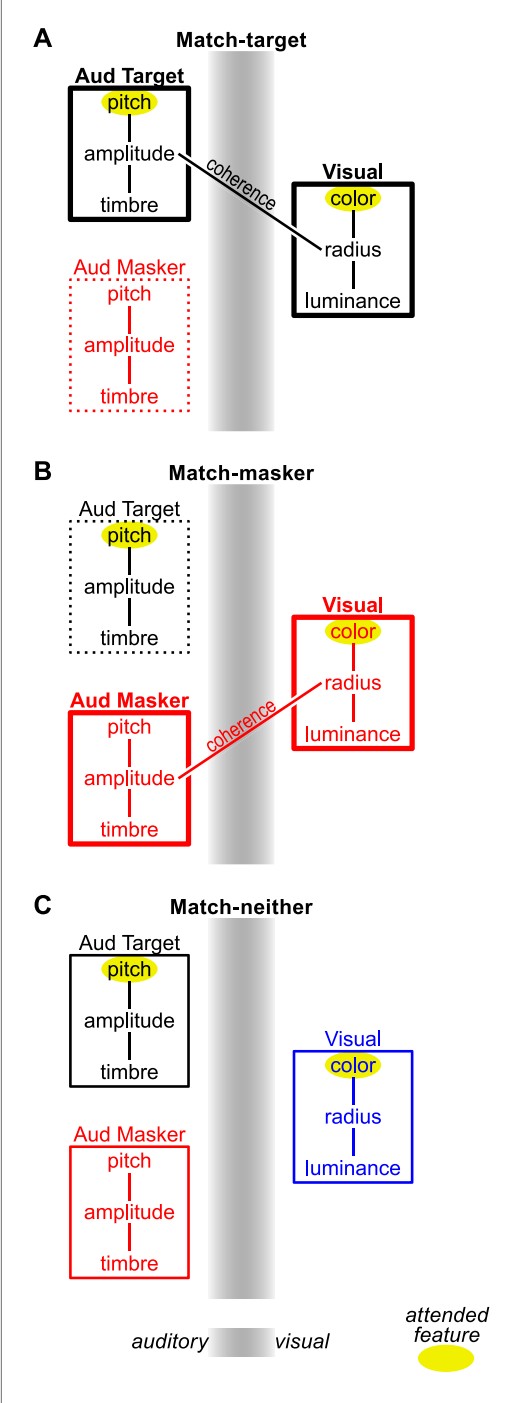

**Figure 3**. Conceptual model of coherence-based cross-modal object formation in the pitch task. Sensory streams are shown as a box containing connected sets of features. Auditory streams are on the left half of the gray sensory boundary and visual on the right. Cross-modal coherence, where present, is shown as a line connecting the coherent auditory and visual features: specifically, the auditory amplitude and the visual size. This results in cross-modal binding of the coherent auditory and visual streams, enhancing each streams'
*Figure 3. Continued on next page*

## Materials and methods

Subjects gave written informed consent and were paid for their participation. Subjects had normal hearing (audiologic thresholds ≤20 dB hearing loss at octave frequencies from 250 to 8000 Hz) and normal or corrected visual acuity. A total of 32 subjects participated (17 male, age 18–33 years). A separate group of 16 subjects each participated in the pitch event and timbre event tasks. The pitch task was run at the University of Washington in the lab of AKCL. The timbre task was performed at the University College London in the lab of JKB.

### General stimulus construction and presentation

Envelopes for visual envelope and auditory amplitude were created using the same frequency domain synthesis. For each trial, an envelope was created by first setting all amplitudes of frequency bins above 0 Hz and below 7 Hz to unity and others to zero. At an audio sampling rate of 24,414 Hz, all non-zero bins were given a random phase from a uniform distribution between 0 and $2\pi$, the corresponding frequency bins across Nyquist frequency were set to the complex conjugates to maintain Hermitian symmetry, and the inverse Fourier transform was computed yielding a time domain envelope. A second and third envelope were created using the same method, and orthogonalized using a Gram-Schmidt procedure. Each envelope was then normalized so that it spanned the interval [0, 1] and then sine-transformed [$y = \sin^2(\pi x/2)$] so that the extremes were slightly accentuated. Visual envelopes were created by subsampling the auditory envelope at the monitor frame-rate of 60 Hz, starting with the first auditory sample, so that auditory amplitude corresponded with the disc radius at the beginning of each frame.

Stimuli were presented in an unlit sound-attenuating room over earphones. Auditory stimuli were created in MATLAB and presented using an RP2 signal processor (Tucker–Davis Technologies, Alachua, FL, USA). Each began and ended with a 10 ms cosine ramp. All stimuli were presented diotically. Visual stimuli were synthesized in MATLAB (The Mathworks, Natick, MA, USA) and presented using the Psychophysics Toolbox (*Brainard, 1997*). Gray discs subtended between 1° and 2.5° at 50 cm viewing distance. The white ring extended 0.125° beyond the gray disc.

Trials lasted 14 s. They began with only the target auditory stimulus and the visual stimulus, indicating the to-be-attended auditory stream to the subject. The auditory masker began 1 s later. As with the rest of the trial, the visual stimulus was only coherent with the auditory target during the

*Figure 3. Continued*

features, which is beneficial in the match-target condition (**A**), problematic in the match-masker condition (**B**), and not present in the match-neither condition (**C**). Attended features are indicated with a yellow ellipse. Enhancement/suppression resulting from object formation is reflected in the strength of the box drawn around each stream's features (i.e., thick lines indicate enhancement, broken lines show suppression).

first second if it was a match-target trial. All streams ended simultaneously. Events did not occur in the first 2 s (1 s after the masker began) or the last 1 s of each trial, or within 1.2 s of any other events in either modality. A response made within 1 s following an event was attributed to that event. Responses not attributed to an event were recorded but were infrequent and are not analyzed here. To ensure audibility and equivalent target to masker ratios without providing confounding information to the subject, an event in either auditory stream or the visual stream could only begin when *both* auditory envelopes were above 75% maximum. There were between 1 and 3 inclusive (mean 2) events in both the target and masker in each trial. There were also between 0 and 2 inclusive (mean 1) visual flashes per trial, in which the outer ring changed from white to cyan (0% red, 100% blue, 100% green) and back. Each subject completed 32 trials of each stimulus condition (96 total), leading to 64 potential hits and 64 potential false alarms for each condition (i.e., 128 responses considered for each *d′* calculation) as well as 32 visual flashes per condition. When computing *d′*, auditory hit and false alarm rates were calculated by adding 0.5 to the numerator and 1 to the denominator so that *d′* had finite limits.

## Pitch task

The pitch task was conducted at the University of Washington. Subjects were seated 50 cm from the screen. Auditory stimuli were presented over earphones (ER-2, Etymotic Research, Elk Grove Village, IL, USA). Each auditory stream was an amplitude-modulated sinusoid with a frequency of either 440 or 565 Hz and was presented at an average of 66 or 63 dB sound pressure level (SPL), respectively in 42 dB SPL white noise to mask any residual sound from outside the sound-treated booth. The higher frequency stream was more salient in piloting and so was attenuated 3 dB so that both streams were of equivalent perceived loudness. Auditory events were 100 ms sinusoidal carrier frequency deflections with a peak amplitude of 1.5 semitones (where *n* semitones is a ratio of $2^{n/12}$; *Figure 1D*).

## Timbre task

The timbre task was conducted at University College London. Subjects were seated 60 cm from the screen with their heads held stationary by a chinrest. Auditory stimuli were presented over headphones (HD 555, Sennheiser, Wedemark, Germany). Each auditory stream was generated as a periodic impulse train and then filtered with synthetic vowels simulated as four-pole filters (formants F1–F4). The /u/ stream (F0 = 175 Hz) had formant peaks F1–F4 at 460, 1105, 2857, 4205 Hz and moved slightly towards /ɛ/ during timbre events, with formant peaks at 730, 2058, 2857, 4205 Hz. The /a/ stream (F0 = 195 Hz) had formant peaks F1–F4 at 936, 1551, 2975, 4263 Hz and moved slightly towards /i/ during timbre events, with formant peaks at 437, 2761, 2975, 4263 Hz. During timbre events the formants moved linearly toward the deviant for 100 ms and then linearly back for 100 ms. Streams were calibrated to be 65 dB SPL (RMS normalized) using an artificial ear (Brüel & Kjær, Nærum, Denmark) and presented against a low level of background noise (54 dB SPL). Before testing, subjects completed a threshold task to determine the size of timbre shift that resulted in 70% detection. The average perturbation (as percentage of distance from steady-state vowel to deviant vowel in the F1–F2 plane) was 12.25%. Four subjects out of twenty tested did not perform well enough to be included (*d′* < 0.7), leading to the final N = 16 analyzed.

## Acknowledgements

We thank Katherine Pratt for assistance with data collection and Dean Pospisil for helpful discussions.

## Additional information

### Funding

| Funder | Grant reference number | Author |
|---|---|---|
| National Institute on Deafness and Other Communication Disorders | R01DC013260 | Adrian KC Lee |
| Wellcome Trust | WT098418MA | Jennifer K Bizley |

| Funder | Grant reference number | Author |
|---|---|---|
| Action on Hearing Loss | 596:UEI:JB | Huriye Atilgan |
| Hearing Health Foundation | Emerging Research Grant | Ross K Maddox |
| Royal Society | International Exchanges Scheme | Jennifer K Bizley, Adrian KC Lee |

The funders had no role in study design, data collection and interpretation, or the decision to submit the work for publication.

## Author contributions

RKM, Conception and design, Acquisition of data, Analysis and interpretation of data, Drafting or revising the article, Contributed unpublished essential data or reagents; HA, Conception and design, Acquisition of data, Analysis and interpretation of data; JKB, AKCL, Conception and design, Analysis and interpretation of data, Drafting or revising the article

## Author ORCIDs

Ross K Maddox, http://orcid.org/0000-0003-2668-0238
Huriye Atilgan, http://orcid.org/0000-0001-8582-4815
Jennifer K Bizley, http://orcid.org/0000-0001-6605-2362
Adrian KC Lee, http://orcid.org/0000-0002-7611-0500

## Ethics

Human subjects: Subjects gave written informed consent and were paid for their participation. All methods were approved by the Institutional Review Board of the University of Washington and the Ethics Committee of the University College London (ref: 5139).

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
