## [Decision Letter]

Thank you for sending your work entitled “Auditory selective attention is enhanced by a task-irrelevant temporally coherent visual stimulus in human listeners” for consideration at *eLife*. Your article has been favorably evaluated by Eve Marder (Senior editor), a Reviewing editor, and 3 reviewers.

The following individuals responsible for the peer review of your submission have agreed to reveal their identity: Jody Culham (Reviewing editor) and Simon Carlile (peer reviewer). Two other reviewers remain anonymous.

The Reviewing editor and the reviewers discussed their comments before we reached this decision, and the Reviewing editor has assembled the following comments to help you prepare a revised submission.

All three reviewers had favorable responses to the manuscript, with remarks such as: “This is an excellent paper which provides original findings that are highly significant”. However, the reviewers brought up points that must be addressed in a revision before the manuscript can be considered suitable for publication in *eLife*.

The reviewing editor has grouped the recommendations into required changes and recommended considerations that have been summarized from individual reviewers’ comments.

1) Two of the reviewers commented that the aim of the paper should be provided earlier in the Introduction.

One reviewer suggests including a statement such as “There has yet to be a study utilizing ongoing, continuous stimuli that demonstrates a perceptual benefit of an uninformative visual stimulus when selectively attending to one auditory stimulus in a mixture”, followed by an overview of previous studies that differed from this goal.

Another reviewer states: “The readability of the manuscript could be improved by a clearer description of the aim of the study, the background literature and the hypotheses addressed. For example, the last sentences of the second paragraph of the Introduction are not clear. What is meant with e.g. “higher-level influences” and “universal feature of multisensory integration”? Also, first the authors write about the influence of visual stimuli on auditory perception, then about the influence of auditory stimuli on visual perception, and then again about the opposite relationship. Please provide better guidance to the reader when discussing these different relationships. In both the Introduction and Discussion sections, the authors repeatedly discuss topics that could be discussed just once. Also, the Discussion section describes studies that provide relevant background, which should be provided in the Introduction section. The conceptual model is very confusing and does not add to the textual description of the results.”

2) There was not much enthusiasm for the first “subjective” experiment. One reviewer thinks it should be deleted. Another reviewer wonders what it really measures. The reviewing editor thought that it got the manuscript off to a weak start and notes that the authors themselves realize this, in the Results section: “The design of this first experiment was admittedly simplistic”.

The authors should “play their strongest card first”—i.e., jump right to the objective experiment. The simplest approach would be to delete the first experiment. However, if the authors do think it is important, they should figure out how to present it in a manner as to not detract from the stronger portions of the paper. If it is included, the following concern from one reviewer should be addressed:

“In the subjective study, subjects were asked to detect binding. This instruction may have resulted in ‘binding experiences’ relatively often, as the subjects were not naïve listeners/viewers. Please discuss this issue.”

Really, though, it doesn't appear that much would be lost by just removing the experiment.

3) The discussion should address the fact that *d'* for the matching target condition did not differ from that in the independent condition. This indicates that binding is only beneficial compared to a condition in which the perception of the masker is facilitated.

4) Alternatives to binding by “objectness” and the framework for considering auditory objects should be discussed.

In the words of one reviewer:

“The main ‘quantitative’ experiment provides strong evidence that the temporal co-modulation of the auditory and visual stimuli does produce a detection advantage for the orthogonal (frequency based) auditory deviations. The overall effect is consistent with the previous literature (although the observation is novel in its own right) and it is very likely that this is mediated by some attentional element in the processing. Indeed, it is plausible that this is mediated through an increase in the ‘objectness’ of the multi-sensory stimulus used here, however, this is not necessarily so. [21] argue that objects will tend to represent actual sources, have boundaries (they can be segregated) and are relatively invariant. These stimuli are artificial so do not reflect the first characteristic and given the data reported here, may reflect the second characteristic. If these features are indeed bound to create a perceptual object then possibly they will reflect invariance. These data may suggest this is the case but the argument is not made explicitly in the manuscript. A more structured frame work for what constitutes an object in general and an auditory object in particular is required and the data discussed in that light.”

Recommended considerations:

1) One reviewer and the Reviewing editor questioned the value added by Figure 4 and recommend removing it.

2) In the analysis, several ANOVAs were conducted. Perhaps a MANOVA is more appropriate, given the dependence of the various dependent measures.

---

## [Author Response]

*The reviewing editor has grouped the recommendations into required changes and recommended considerations that have been summarized from individual reviewers’ comments*.

We appreciate the reviewers’ constructive criticism. We have made all of the required revisions, and the majority of the requested revisions (the model figure was retained, but made less stylized). We believe these changes make the manuscript clearer, more concise and direct, focusing on the most novel and important results and discussion.

*1) Two of the reviewers commented that the aim of the paper should be provided earlier in the Introduction*.

*One reviewer suggests including a statement such as “There has yet to be a study utilizing ongoing, continuous stimuli that demonstrates a perceptual benefit of an uninformative visual stimulus when selectively attending to one auditory stimulus in a mixture”, followed by an overview of previous studies that differed from this goal*.

We agree that making the purpose of the study clear sooner in the Introduction will strengthen it. We have taken the quoted sentence and moved it to the end of the first paragraph, motivating the remainder of the Introduction.

*Another reviewer states: “The readability of the manuscript could be improved by a clearer description of the aim of the study, the background literature and the hypotheses addressed. For example, the last sentences of the second paragraph of the Introduction are not clear. What is meant with e.g. “higher-level influences” and universal feature of multisensory integration”? Also, first the authors write about the influence of visual stimuli on auditory perception, then about the influence of auditory stimuli on visual perception, and then again about the opposite relationship. Please provide better guidance to the reader when discussing these different relationships. In both the Introduction and Discussion sections, the authors repeatedly discuss topics that could be discussed just once. Also, the Discussion section describes studies that provide relevant background which should be provided in the Introduction section. The conceptual model is very confusing and does not add to the textual description of the results*.*”*

Based on these suggestions and in those of the previous reviewer, we have made a number of changes to the Introduction and Discussion sections that we hope address these concerns, moving some important concepts to the Introduction (“Past studies employing transient stimuli…” were moved from the Discussion to the Introduction) and drastically reducing the redundancy with which some topics were discussed (“Many studies have examined how…” represent two combined paragraphs that were previously in the Introduction and Discussion).

*2) There was not much enthusiasm for the first “subjective” experiment. One reviewer thinks it should be deleted. Another reviewer wonders what it really measures. The reviewing editor thought that it got the manuscript off to a weak start and notes that the authors themselves realize this, in the Results section: “The design of this first experiment was admittedly simplistic”*.

*The authors should “play their strongest card first”—i.e., jump right to the objective experiment. The simplest approach would be to delete the first experiment. However, if the authors do think it is important, they should figure out how to present it in a manner as to not detract from the stronger portions of the paper. If it is included, the following concern from one reviewer should be addressed*:

*“In the subjective study, subjects were asked to detect binding. This instruction may have resulted in ‘binding experiences’ relatively often, as the subjects were not naïve listeners/viewers. Please discuss this issue*.*”*

*Really, though, it doesn't appear that much would be lost by just removing the experiment*.

We agree with the reviewers that perhaps including the experiment, while it does provide some information, makes the manuscript less effective overall. We have thus removed the subjective experiment and the corresponding figure 1.

*3) The discussion should address the fact that* d' *for the matching target condition did not differ from that in the independent condition. This indicates that binding is only beneficial compared to a condition in which the perception of the masker is facilitated.*

This point is now addressed in the Results section where we address the basic trends found in the results (“There was no significant difference between…”).

*4) Alternatives to binding by “objectness” and the framework for considering auditory objects should be discussed*.

*In the words of one reviewer*:

*“The main ‘quantitative’ experiment provides strong evidence that the temporal co-modulation of the auditory and visual stimuli does produce a detection advantage for the orthogonal (frequency based) auditory deviations. The overall effect is consistent with the previous literature (although the observation is novel in its own right) and it is very likely that this is mediated by some attentional element in the processing. Indeed, it is plausible that this is mediated through an increase in the ‘objectness’ of the multi-sensory stimulus used here, however, this is not necessarily so.*
[21]
*argue that objects will tend to represent actual sources, have boundaries (they can be segregated) and are relatively invariant. These stimuli are artificial so do not reflect the first characteristic and given the data reported here, may reflect the second characteristic. If these features are indeed bound to create a perceptual object then possibly they will reflect invariance. These data may suggest this is the case but the argument is not made explicitly in the manuscript. A more structured frame work for what constitutes an object in general and an auditory object in particular is required and the data discussed in that light*.*”*

We now discuss the Griffiths and Warren paper in depth in the Discussion section (“Griffiths and Warren [37] suggest…”), which provides a set of criteria for what could define an auditory object. We then use this as a launching point for a discussion on the perceptual *benefits* of auditory-visual object formation, and how measuring such benefits actually provide an objective test of whether an object was formed or not.

*Recommended considerations*:

*1) One reviewer and the Reviewing editor questioned the value added by Figure 4 and recommend removing it*.

We included the original Figure 4 to provide a visual aid for what we feel is a complicated topic. Having shown the figure to a few people locally, some seemed not to like the rather stylized look of it. We have hence opted to retain it (now as Figure 3), but have reworked it with a more modest aesthetic.

*2) In the analysis, several ANOVAs were conducted. Perhaps a MANOVA is more appropriate, given the dependence of the various dependent measures*.

While *d′* and bias are each dependent on hit rate and false alarm rate, there is no dependence between *d′* and bias, or between false alarm rate and hit rate, and none of these should be dependent with visual hit rate. While the MANOVA might provide a bit more statistical power, we feel that the separate ANOVAs run here are easier to interpret while being more specific.